# Molecular Variation in Some Taxa of Genus *Astragalus* L. (Fabaceae) in the Iraqi Kurdistan Region

Lanja Hewa Khal, Nawroz Abdul-razzak Tahir * 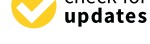 and Rupak Tofiq Abdul-Razaq

Department of Horticulture, College of Agricultural Engineering Sciences, University of Sulaimani, Sulaimani 46001, Iraq; lanja.khall@univsul.edu.iq (L.H.K.); rupak.abdulrazaq@univsul.edu.iq (R.T.A.-R.)
* Correspondence: nawroz.tahir@univsul.edu.iq; Tel.: +964-7701965517

**Abstract:** *Astragalus* L. is one of the main genera of blossoming plants, and its diversity of species and forms is well known. The *Astragalus* L. taxa make a significant contribution to the mountainous and steppe ecosystems of the Kurdistan region of Iraq. Although the species within this genus have been the subject of several molecular studies, the evolutionary relationships among these species remain unidentified. Despite extensive research, little is known about the phylogenetic relationships among the various subgenera of *Astragalus*. This research is intended to investigate the molecular variation of 33 species of the genus *Astragalus* L. found in the Kurdistan area of Iraq. For molecular validations, three separate techniques (nuclear ribosomal DNA (rDNA), inter simple sequence repeats (ISSR), and conserved DNA-derived polymorphism (CDDP)) were used. With respect to *Astragalus* L. indentations, universal ITS1 and ITS4 gene sequencing was used, and the discovered sequences were subjected to BLAST searches in the NCBI database. A phylogenetic tree was generated with two main clades. Regarding detecting genetic diversity between the taxa, 24 molecular markers (14 ISSR and 10 CDDP) were used. High values of polymorphic materials and gene diversity were detected. ISSR markers had an average of 22.71 polymorphic bands per primer, while CDDP markers had the highest mean values for polymorphic information content (0.37), Shannon's information index (0.27), expected heterozygosity (0.19), and unbiased expected heterozygosity (0.23). Cluster and principal coordinate analyses divided the *Astragalus* L. taxa into four main groups using the two molecular data sets. According to the results of the molecular variance analysis, the highest variation was detected within sections, with values of 92.01 and 89.48% for ISSR and CDDP markers, respectively. These outcomes suggest the effectiveness of molecular markers and the ITS region in determining and identifying genetic correlations between *Astragalus* species.

**Keywords:** flowering plant; flora of Kurdistan; molecular study; classification; genetic diversity



## 1. Introduction

*Astragalus* L. is a member of the Fabaceae plant family and the Papilionoideae subfamily of the Astragalinae subtribe of the Galegeae tribe, with significant medicinal and economic importance. *Astragalus* is widely recognized to have antiviral, anti-inflammatory, innate immune priming, and NLRP3-mediated inflammation-reducing properties. Furthermore, the swainsonine alkaloid in these plants blocks the glycosylation required for the binding of SARS-CoV-2 to human cells [1]. After Orchidaceae and Asteraceae, Fabaceae is the third biggest flowering plant family in the world [2,3]. With over 2400 species, it is one of the world's biggest genera of vascular plants [4]. Geographical differences have an obvious influence on the morphological and even anatomical properties of the plant, as well as how genes interact with environmental factors to produce the phenotypic traits of the particular plant [5,6]. Environmental factors have a significant impact on influencing the prevalence and distribution of diverse species, whether on a global or country-by-country scale [7]. For identifying purposes, the whole plant had to be picked at the same time of year when it was growing, flowering, and making fruit. Only trained taxonomists were able

to figure out the species correctly. These processes are hard, take extensive time, and may not be able to correctly recognize these plants at the species level because these traits often change from the same species of ancestor, and are intensely changed by environmental or growing factors during a plant's development [8]. Even a skilled taxonomist might get it wrong if the taxa have the same phenotype or were collected when they were still young. Due to taxonomic problems between the complex species of the genus *Astragalus* [9], these issues still need to be looked at on a molecular scale to confirm their taxonomic identity, the partnership within and between them, and the use of phylogenetic trees to understand their genetic differences.

Making use of morphological data and several genetic markers like chloroplast DNA (cpDNA) and nrDNA, numerous researchers have investigated the phylogenetic correlations within the *Astragalus* genus [5,10,11]. Hence, the boundaries of *Astragalus* and its phylogenetic placement have been a source of contention for a long time. Most authors reported that the ITS area (ITS1; 5.8-S; ITS2 subunits) of 18S–26S nrDNA is often desired in molecular works [12]. The range of diversity in this area makes it appropriate for phylogenetic inference at the species, genus, and family levels [13,14]. Both ISSR and CDDP markers can survey a large number of loci with a small amount of plant material, making them more suitable techniques. They are cheap and efficient molecular methods for detecting polymorphisms in plants. ISSR markers can be used to target several regions of the genome with minimal effort and expense by amplifying inter-microsatellite sequences with polymerase chain reactions [15]. CDDP molecular markers are a cutting-edge method that amplify the entire plant genome with primers based on conserved DNA sequences, producing bands and targets mostly related to biotic and abiotic stress. Many molecular markers, such as ISSR [15,16] and SRAP [17], have recently been used to examine phylogenetic relationships in *Astragalus* species. Aside from the utility of the previously mentioned markers, a new form of marker known as CDDP has emerged as a valuable tool for measuring diversity in various plant species [18–20]. There have been considerable efforts in Turkey [21–23] and Iran [16,24] regarding the morphological diversity hub for *Astragalus* species.

There is a lack of or no knowledge of the taxonomic profile and genetic diversity of *Astragalus* species in Iraq. *Astragalus* species in the Kurdistan region of Iraq have been declining due to long-term land use and rapid construction; these problems may become more serious due to climate change. To certify the longevity of *Astragalus* species, it is necessary to take an integrated approach, which includes field surveys for genetic diversity, cultivation, and natural habitat restoration. This work represents the first comprehensive taxonomic and structural study of Iraqi *Astragalus* species using CDDP markers, and to the best of our knowledge, no previous studies have used these markers to examine *Astragalus* species elsewhere. Thus, the current study intends to use ITS, ISSR, and CDDP markers to determine the relationship among 33 *Astragalus* species in the Iraqi Kurdistan region. A molecular classification of *Astragalus* species in Iraqi Kurdistan would be a significant addition to understanding the evolutionary history of *Astragalus* species.

## 2. Materials and Methods

### 2.1. Taxon Sampling

In total, 33 species of the genus *Astragalus* L. accessions were gathered from the survey started from February 2021 to July 2023 in different parts of the Kurdistan region of Iraq, including Sulaimaniyah (MSU), Erbil (FAR), Kirkuk (FKI), Rouandwz (MRO), and Amadya provinces (MAM), from their natural habitats (Table 1 and Figure 1). On the field, species were gathered based on their morphological characteristics, including their roots, stems, leaves, flowers, fruits, and seeds [25,26]. Important morphological characteristics such as the type of stem (erect, cushion, prostrate, short spreading stem, and ascending), the stipules (type of shape, dimensions, apex, and indumentum), and the characteristics of compound leaves were used in the identification process (length of leaves, number of leaflets pairs, dimensions and shape of leaflets, length leaf rachides, apex of leaflets and indumentum). The characteristics of the flowering system include the quantity and type of

flowers; the size of the peduncle and pedicle; the presence of bracts and bracteolates; the length, shape, and indument of the calyx and tooth; the dimensions and shape of the banner, wings, and keel; the androecium and gynoecium; and the fruit (including the size, shape, and indument of the pod, and the length). Following collection, these were thoroughly examined by taxonomists and compared to specimens in the national herbarium of Iraq as well as the flora of Iraq and its neighbors. The vouchers were designated with numbers AL-1 to AL-33. To keep the leaf samples fresh, they were plucked and crushed using liquid nitrogen before being transported to the lab in the field. The samples were removed from the container and kept frozen until they were used.

**Table 1.** The list of *Astragalus* L. species and their accession numbers obtained from rDNA ITS sequences.

| Section Name | Taxa Name | Voucher Number | Location | Altitude (m) | Accession Number |
|---|---|---|---|---|---|
| Adiaspastus (Sect-1) | *Astragalus caspicus* M. Bieb. | AL-1 | MSU | 1570 | OP947045 |
| | *A. michauxianum* Boiss. | AL-2 | MSU, MRO, and MAM | 1296–2070 | OP964729 |
| Alopecias (Sect-2) | *A. echinops* Auch. ex Boiss. | AL-3 | MSU and MRO | 860–1380 | OP947536 |
| | *A. macrocephalus* subsp. *Finitimus* Bunge. | AL-4 | MSU and MRO | 1210–1400 | OP948301 |
| Hymenostegis (Sect-3) | *A. obtusifolius* DC. | AL-5 | MSU, MRO, and MAM | 160–340 | OP950215 |
| | *A. chrysostachys* Boiss. | AL-6 | MSU | 1350–1790 | OP947139 |
| | *A. glumaceus* Boiss. | AL-7 | MSU | 1735 | OP948166 |
| | *A. lagurus* Willd. | AL-8 | MSU and MAM | 1330–1680 | OP959921 |
| | *A. nervistipulus* Boiss. & Hausskn. | AL-9 | MSU | 1360–1750 | OP948699 |
| Incai (Sect-4) | *A. cyclophyllon* Beck. | AL-10 | MSU | 1090 | OP947523 |
| | *A. latifolius* Lam. | AL-11 | MSU | 1247 | OP948257 |
| Macrophyllium (Sect-5) | *A. basianicus* Boiss. & Hausskn. | AL-12 | MSU and MRO | 870–1490 | OP964679 |
| | *A. compactus* Lam. | AL-13 | MSU | 1387 | OP947498 |
| Malacothrix (Sect-6) | *A. iranicus* Bunge. | AL-14 | MSU, MRO, FAR, and MAM | 900–1970 | OP948645 |
| | *A. sarae* Eig. | AL-15 | MSU and MAM | 1100–1450 | OP964723 |
| | *A. singarensis* Boiss. | AL-16 | MSU, MRO and MAM | 800–1725 | OP964726 |
| | *A. spachianus* Boiss. & Buhse. | AL-17 | MSU | 1495 | OP984381 |
| Myobroma (Sect-7) | *A. aegobromus* Boiss. | AL-18 | MSU and MRO | 1300–2450 | OP942443 |
| | *A. brachystachys* DC. | AL-19 | MSU, MRO, and MAM | 250–1200 | OP964725 |
| | *A. lobophorus* Boiss. | AL-20 | MSU, MRO and MAM | 1000–1680 | OP948282 |
| | *A. macropelmatus* Bunge. | AL-21 | MSU and MRO | 1360–1840 | OP948644 |
| | *A. pinetorum* subsp. *Declinatus* (Kuntze.) Podlech. | AL-22 | MSU and FAR | 1980–2795 | OP950280 |
| | *A. rawianus* C.C. Towns. | AL-23 | MSU and MAM | 1480–2055 | OP964724 |
| Onobrychium (Sect-8) | *A. aduncus* Willd. | AL-24 | MSU and MAM | 870 | OP942435 |
| | *A. karamasicus* Boiss. & Balansa. | AL-25 | MSU | 1200 | OP948228 |
| | *A. jacobsii* Podlech. | AL-26 | MSU | 997 | OP948225 |
| Poterion (Sect-9) | *A. fasciculifolius* Boiss. | AL-27 | MSU and FKI | 250–470 | OP957370 |
| | *A. spinosus* (Forssk.) Muschl. | AL-28 | MSU | 435–660 | OP984382 |
| Proseliu (Sect-10) | *A. curvirostris* Boiss. | AL-29 | MSU | 1160–1400 | OP984362 |
| | *A. micrancistrus* Boiss. & Hausskn. | AL-30 | MSU and MRO | 1420–1870 | OP950200 |
| Rhacophorus (Sect-11) | *A. cephalotes* Banks & Sol. | AL-31 | MSU, MRO, and MAM | 720–1280 | OP947089 |
| | *A. globiflorus* Boiss. | AL-32 | MSU | 1400–1680 | OP948150 |
| | *A. microcephalus* Willd. | AL-33 | MSU and MRO | 1480–2370 | OP950203 |

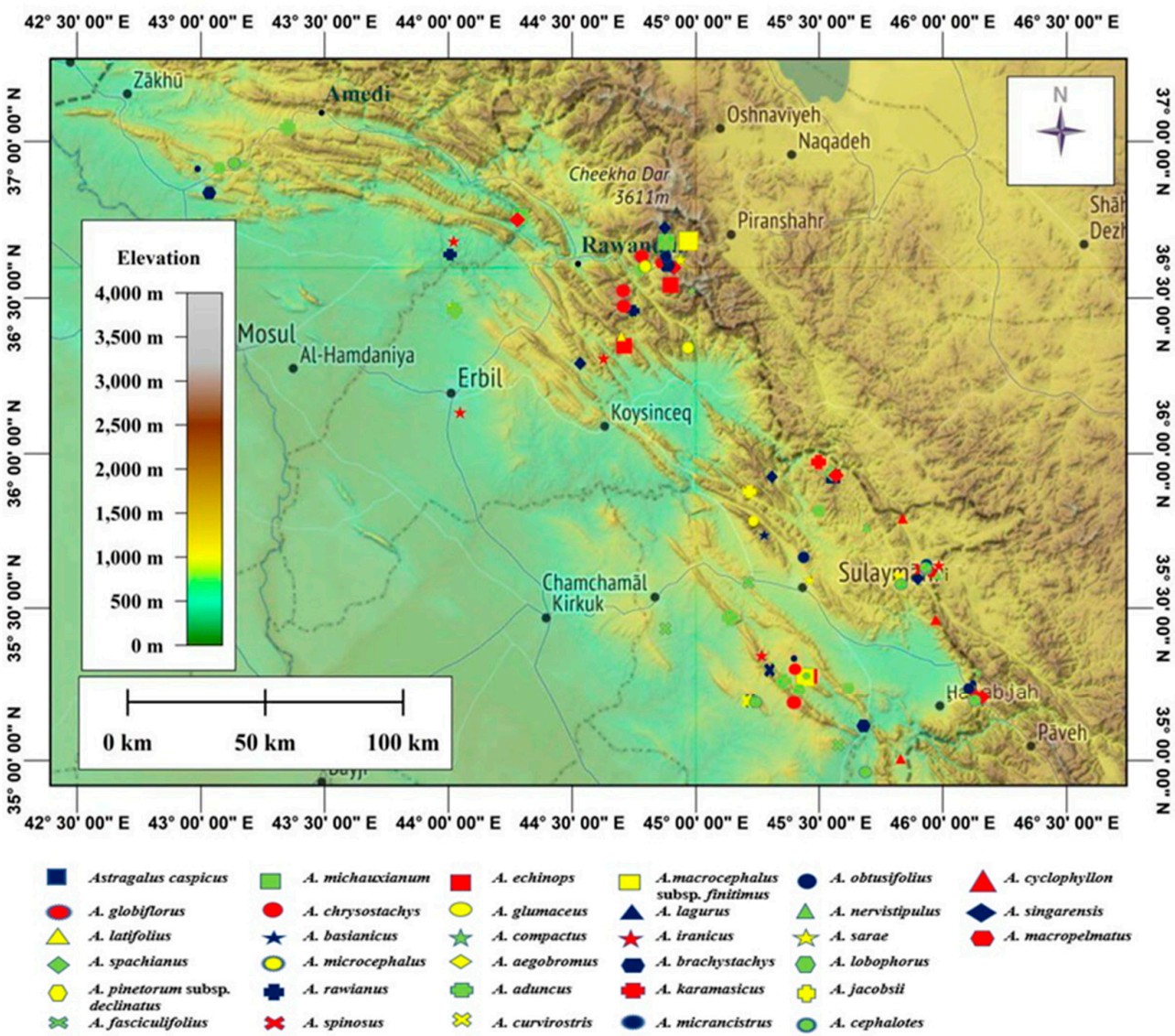

**Figure 1.** A map showing the sample location of different species of *Astragalus* L. in Iraqi Kurdistan.

### 2.2. DNA Extraction

For DNA extraction, 300 mg of powdered leaf material was taken from each sample. A nearly identical protocol to that described by Ahmed et al. [18] was chosen, with the only difference being that the lysis time was increased to 90 min at 65 °C rather than 75 min at 63 °C. Agarose gel (1.1%) and a nanodrop spectrophotometer (Nano PLUS, MAANLAB AB, Växjö, Sweden) were used to evaluate the quality and quantity of isolated DNA.

### 2.3. PCR Assay, Purification, and Sequencing

For amplification of the ITS region, ITS1 (5′ TCCGTAGGTGAACCTGCGG 3′) and ITS4 (5′ TCCTCCGCTTATTGATATGC 3′) were employed [27]. In brief, the PCR amplification was carried out in a 26 μL volume that contained 5.0 μL of genomic DNA, 10 μL of master mix (AddStart Taq Master, AddBio, Daejeon, Republic of Korea), 2 μL of each ITS primer (10 μM), and 7 μL of deionized water. The first step of the PCR program was to separate the strands at 94 °C for 10 min, followed by 35 cycles of denaturation at 94 °C for 1 min, annealing at 55 °C for 1 min, elongation at 72 °C for 2 min, and a final extension at 72 °C for 10 min. A negative control (no DNA template) was utilized to monitor each set of reactions. Following PCR amplification, the product was separated on 1.5% agarose gels with 1X TBE buffer, stained with ethidium bromide, and observed under UV light. The ITS bands

were cut, and the DNA was extracted from the agarose using a Gel Extraction Kit (AddBio, Daejeon, Republic of Korea). The DNA was then sequenced by the Macrogen Company in South Korea, and the results were compared to DNA sequences previously recorded in the National Center for Biotechnology Information dataset (NCBI).

Regarding the PCR procedure of ISSR and CDDP markers, a total of 24 different markers (14 primers for ISSR and 10 primers for CDDP) were used (Table 2) [19,20,28]. A similar PCR procedure, as stated previously, was used with the only exception of changing the annealing temperature with regard to using different molecular markers. The bands were separated on an agarose gel (1.8%) stained with ethidium bromide and detected under UV light.

**Table 2.** Name and sequence of different ISSR and CDDP markers used in the assessment of *Astragalus* L. taxa.

| Markers | Annealing Temperature (°C) | Sequence (5′–3′) |
| --- | --- | --- |
| ISSR | | |
| ISSR-6 | 50 | GCCTCCTCCTCCTCCTCC |
| ISSR-9 | 50 | CACACACACACACACATG |
| ISSR-11 | 50 | ACACACACACACACACGG |
| ISSR-12 | 50 | AGAGAGAGAGAGAGAGCT |
| UBC-810 | 50 | GAGAGAGAGAGAGAGAT |
| UBC-814 | 50 | CTCTCTCTCTCTCTCTA |
| UBC-815 | 50 | CTCTCTCTCTCTCTCTG |
| UBC-818 | 50 | CACACACACACACACAG |
| UBC-822 | 50 | TCTCTCTCTCTCTCTCA |
| UBC-825 | 50 | ACACACACACACACACT |
| UBC-826 | 50 | ACACACACACACACACC |
| UBC-834 | 50 | AGAGAGAGAGAGAGAGGT |
| UBC-841 | 50 | GAGAGAGAGAGAGAGACTC |
| UBC-847 | 50 | CACACACACACACACAGC |
| CDDP | | |
| ABP-1 | 50 | ACSCCSATCCACCGC |
| ERF=1 | 50 | CACTACCCCGGSCTSCG |
| ERF-2 | 50 | GCSGAGATCCGSGACCC |
| KNOX-2 | 50 | CACTGGTGGGAGCTSCAC |
| KNOX-3 | 50 | AAGCGSCACTGGAAGCC |
| MYB-1 | 50 | GGCAAGGGCTGCCGC |
| MYB-2 | 50 | GGCAAGGGCTGCCGG |
| WRKY-R1 | 50 | GTGGTTGTGTCTTGCC |
| WRKY-R3 | 50 | GCASGTGTGCTCGCC |
| WRKY-R3B | 50 | CCGCTCGTGTGSACG |

*2.4. Phylogenetic Tree Development*

In order to obtain accurate sequences of a certain region, the DNA was sequenced in both directions. With SnapGene version 3.1 software, chromatograms were used to double-check sequences, and sequences were re-run when necessary. The sequences were aligned using the Clustal W tool of Molecular Evolutionary Genetics Analysis software (MEGA 11). Using the neighbor-joining tree method, a phylogenetic tree was constructed with 1000 bootstrap replicates to ensure pre-trial reliability [29,30].

*2.5. Scoring and Statistical Analysis*

The visible PCR products obtained with ISSR and CDDP were scored. To reduce the likelihood of error, only distinct, reproducible, and dense bands were assessed. To build a binary matrix for each sample, the presence of amplified bands was represented by a value of 1, while the absence of a band was represented by 0. To obtain the Jaccard similarity coefficient, the scored data matrices were analyzed statistically with XLSTAT version 2020.1.3. The cluster analysis was carried out using an unweighted pair-group

procedure with arithmetic averages (UPGMA) for creating the dendrogram tree. Using XLSTAT 2020.1.3, the principal coordinate analysis was carried out. The polymorphism information content (PIC), genetic distance, diversity indices, and analysis of molecular variance (AMOVA) were calculated using GenAIEx version 6.51b2.

## 3. Results

### 3.1. Alignment of Sequences and Phylogenetic Reconstruction of the Genus Astragalus L.

The amplified ITS rDNA PCR product was sequenced and compared to the public database using BLAST. All taxa's ITS sequences were checked, and new accessions were added to NCBI GenBank (Table 1). The sequence of the tested species ranged from 601–643 bp, according to a molecular analysis of 33 taxa of the genus *Astragalus* L. using ITS sequencing with universal primer pairs ITS1 and ITS4. The sequences contained 18S rRNA, the entire ITS1, and 5.8S rRNA regions. The complete set of *Astragalus* L. species was used to generate a neighbor-joining tree through a series of nucleotide sequence alignments. The tree divided 33 taxa into two clades (Figure 2). Group 1 (Gr-1) consists of five sub-clades (1–5). Sub-clade 1 consists of four taxa, which are *A. curvirostris, A. latifolius, A. cyclophyllon,* and *A. micrancistrus.* The relationship between them was 72%, 72%, and 50%, respectively, in the same cluster. While sub-clade 2 consists of only *A. pinetorum* subsp. *declinatus,* the similarity of these sub-clades with sub-clade 3 was 86%. Five taxa were aligned in sub-clade 3: *A. brachystachys, A. aegobromus, A. rawianus, A. chrysostachys,* and *A. sarae,* with a similarity of 81%, 81%, 84%, and 66%, respectively. The only taxon that separated from the rest of the studied taxa was *A. macropelmatus* in sub-clade 4, and the similarity of these sub-clades with sub-clade 3 was 91%. The relationships between *A. michauxianum, A. macrocephalus* subsp. *finitimus, A. obtusifolius,* and *A. echinops* in sub-clade 5 were 98%, 97%, and 96%, respectively. Group 2 (Gr-2), on the other hand, consists of six sub-clades. Sub-clade 1 consists of only one taxon, *A. spachianus;* the similarity of these sub-clades with sub-clade 2 was 88%. The sub-clade 2 consists of five taxa, which are *A. lobophorus, A. iranicus, A. jacobsii, A. karamasicus,* and *A. aduncus,* and the relationship between them is 23%, 65%, 51%, and 44%, respectively. Also, *A. microcephalus* was separated alone in sub-clade 3, and shared relativeness with sub-clade 4 by 79%. *A. glumaceus, A. caspicus, A. lagurus, A. singarensis,* and *A. nervistipulus* were placed in sub-clade 4, and their similarity was 30%, 69%, 37%, and 72%, respectively. Sub-clade 5 consists of five taxa: *A. basianicus, A. globiflorus, A. spinosus, A. cephalotes,* and *A. compactus.* Finally, sub-clade 6 consists of only *A. fasciculifolius,* and these two sub-clades are related by 88%. These results revealed that there were discernible divisions in the molecular phylogeny of the various taxonomic forms of *Astragalus.* An examination of the DNA sequence derived from both ITS regions revealed a significant amount of genetic diversity among the samples that were studied.

### 3.2. Importance of Using ISSR Markers in Detecting Genetic Diversity

All of the ISSR primers generated scoreable and clearly defined amplification products, and they revealed polymorphisms among the investigated 33 *Astragalus* taxonomic groups. The 14 ISSR primers produced 318 polymorphic bands. The polymorphic bands ranged between 17 (UBC-847) and 28 bands (ISSR-9 and ISSR-11). According to the findings, the polymorphic information content (PIC) values ranged from 0.30 (ISSR-9) to 0.40 (UBC-814), with a mean of 0.35. The different allele numbers (Na), effective allele numbers (Ne), Shannon's information index (I), expected heterozygosity (He), and expected unbiased heterozygosity (uHe) ranged from 0.83 to 1.26, 1.22 to 1.30, 0.20 to 0.31, 0.13 to 0.21, and 0.16 to 0.26, respectively (Table 3).

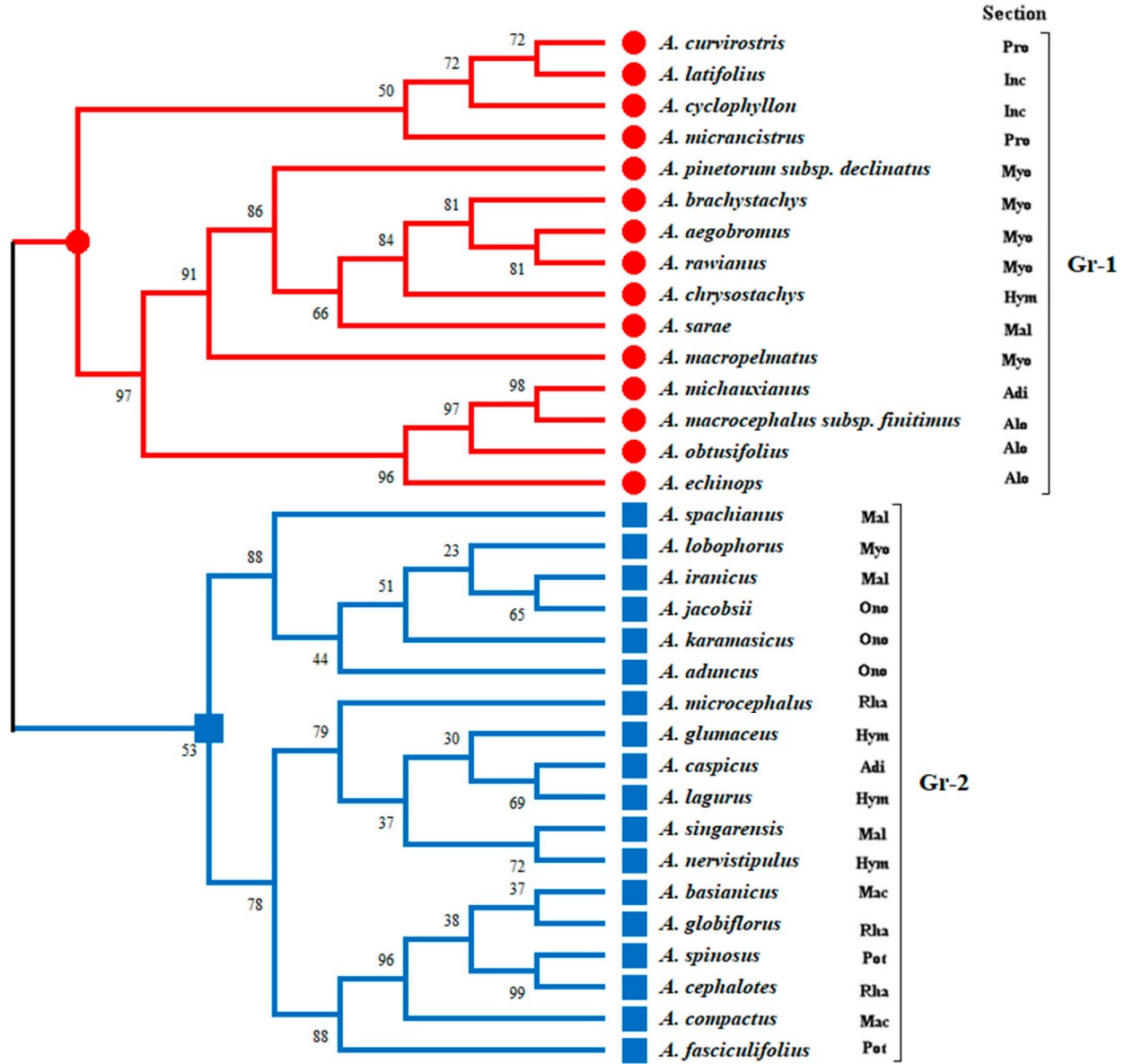

**Figure 2.** Neighbor-joining tree for 33 *Astragalus* L. taxa using MEGA 11 as the nucleotide substitution model. The branch lengths are relative to the number of predicted nucleotide substitutions. The numbers represent the bootstrap values. Different colors signify various groups (Gr).

**Table 3.** Polymorphic efficacy and diversity indices of ISSR primers detected in the taxa of the genus *Astragalus* L.

| ISSR Markers | NPB | PIC | Na | Ne | I | He | uHe |
|---|---|---|---|---|---|---|---|
| ISSR-6 | 24.00 | 0.33 | 0.99 | 1.28 | 0.23 | 0.16 | 0.19 |
| ISSR-9 | 28.00 | 0.30 | 0.83 | 1.22 | 0.20 | 0.13 | 0.16 |
| ISSR-11 | 28.00 | 0.32 | 0.97 | 1.27 | 0.24 | 0.16 | 0.20 |
| ISSR-12 | 22.00 | 0.37 | 1.06 | 1.30 | 0.27 | 0.18 | 0.22 |
| UBC-810 | 21.00 | 0.37 | 1.03 | 1.28 | 0.26 | 0.17 | 0.21 |
| UBC-818 | 20.00 | 0.37 | 1.07 | 1.30 | 0.27 | 0.18 | 0.22 |
| UBC-814 | 23.00 | 0.40 | 1.26 | 1.35 | 0.31 | 0.21 | 0.26 |
| UBC-815 | 25.00 | 0.38 | 1.21 | 1.34 | 0.31 | 0.21 | 0.25 |
| UBC-822 | 24.00 | 0.37 | 1.05 | 1.30 | 0.26 | 0.17 | 0.21 |

**Table 3.** *Cont.*

| ISSR Markers | NPB | PIC | Na | Ne | I | He | uHe |
|---|---|---|---|---|---|---|---|
| UBC-825 | 20.00 | 0.36 | 0.93 | 1.26 | 0.23 | 0.15 | 0.19 |
| UBC-826 | 25.00 | 0.32 | 0.89 | 1.25 | 0.22 | 0.15 | 0.18 |
| UBC-834 | 18.00 | 0.31 | 0.89 | 1.27 | 0.23 | 0.16 | 0.19 |
| UBC-841 | 23.00 | 0.33 | 0.98 | 1.29 | 0.25 | 0.17 | 0.21 |
| UBC-847 | 17.00 | 0.31 | 0.99 | 1.27 | 0.23 | 0.16 | 0.19 |
| Total | 318.00 | 4.84 | 14.15 | 17.96 | 3.51 | 2.36 | 2.89 |
| Mean | 22.71 | 0.35 | 1.01 | 1.28 | 0.25 | 0.17 | 0.21 |

NPB: number of polymorphic bands, PIC: polymorphic information content, Na: number of different alleles, Ne: number of effective alleles, I: Shannon's information index, He: expected heterozygosity, uHe: unbiased expected heterozygosity.

### 3.3. Polymorphism Parameters of CDDP Markers

Scoreable fragments were produced by CDDP primers. Some 132 polymorphic bands were amplified across all taxa of the genus *Astragalus* L., with a 13.2 band average per primer. KNOX-2 (16 bands) and WRKY-R3 (10 bands) had the most/least polymorphic bands. The PIC values in the CDDP primers were ranked from lowest to highest as follows: WRKY-R3 < KNOX-3 < ERF-1 < KNOX-2 and MYB-1 < ERF-2, MYB-2 and WRKY-R3B < ABP-1. The Na, Ne, I, He, and uHe levels were 0.85 to 1.29, 1.23 to 1.42, 0.19 to 0.35, 0.13 to 0.24, and 0.17 to 0.30, respectively (Table 4).

**Table 4.** Polymorphic effectiveness and diversity indices of CDDP primers found in the taxa of the genus *Astragalus* L.

| CDDP Markers | NPB | PIC | Na | Ne | I | He | uHe |
|---|---|---|---|---|---|---|---|
| ABP-1 | 15.00 | 0.41 | 1.15 | 1.31 | 0.27 | 0.18 | 0.22 |
| ERF-1 | 15.00 | 0.33 | 0.98 | 1.28 | 0.24 | 0.16 | 0.20 |
| ERF-2 | 14.00 | 0.40 | 1.32 | 1.42 | 0.35 | 0.24 | 0.30 |
| KNOX-2 | 16.00 | 0.37 | 0.99 | 1.29 | 0.24 | 0.16 | 0.20 |
| KNOX-3 | 12.00 | 0.31 | 0.92 | 1.23 | 0.19 | 0.13 | 0.16 |
| MYB-1 | 12.00 | 0.37 | 1.17 | 1.35 | 0.29 | 0.20 | 0.25 |
| MYB-2 | 13.00 | 0.40 | 1.07 | 1.33 | 0.28 | 0.19 | 0.23 |
| WRKY-R1 | 13.00 | 0.44 | 1.25 | 1.40 | 0.34 | 0.23 | 0.28 |
| WRKY-R3 | 10.00 | 0.29 | 0.85 | 1.24 | 0.20 | 0.14 | 0.17 |
| WRKY-R3B | 12.00 | 0.40 | 1.29 | 1.38 | 0.31 | 0.22 | 0.26 |
| Total | 132.00 | 3.72 | 10.98 | 13.22 | 2.72 | 1.85 | 2.26 |
| Mean | 13.20 | 0.37 | 1.10 | 1.32 | 0.27 | 0.19 | 0.23 |

NPB: number of polymorphic bands, PIC: polymorphic information content, Na: number of different alleles, Ne: number of effective alleles, I: Shannon's information index, He: expected heterozygosity, uHe: unbiased expected heterozygosity.

### 3.4. Cluster Analysis of Astragalus L. Taxa

In the dendrogram that was constructed based on the ISSR data, our 33 taxonomic species were split into four groups (Figure 3A). The first group (G-1) had two different genetically similar taxa. The second (G-2) was made up of three taxa. The third group (G-3) consisted of 17 taxa and the last group (G-4) contained 11 taxa. As a result of the phylogenetic grouping of the dissimilarity matrix (Figure 3B), the CDDP analysis resulted in the production of a dendrogram that featured four primary clusters. The first group (G-1) consisted of nine taxa, which subdivided into two subgroups. The first subgroup includes four taxa, while the second subgroup contains five taxa. The second group (G-2) contained three taxa, which divided into two subgroups. The third group (G-3) consisted of 11 taxa, which contained two subgroups, and the last group (G-4) contained 10 taxa, which were classified into two subgroups. The dendrogram shows that the molecular data sets can distinguish between *Astragalus* taxa based on their origin region. Furthermore, the data

revealed a direct link between the molecular clusters, the origin of the accessions, and the source of those accessions.

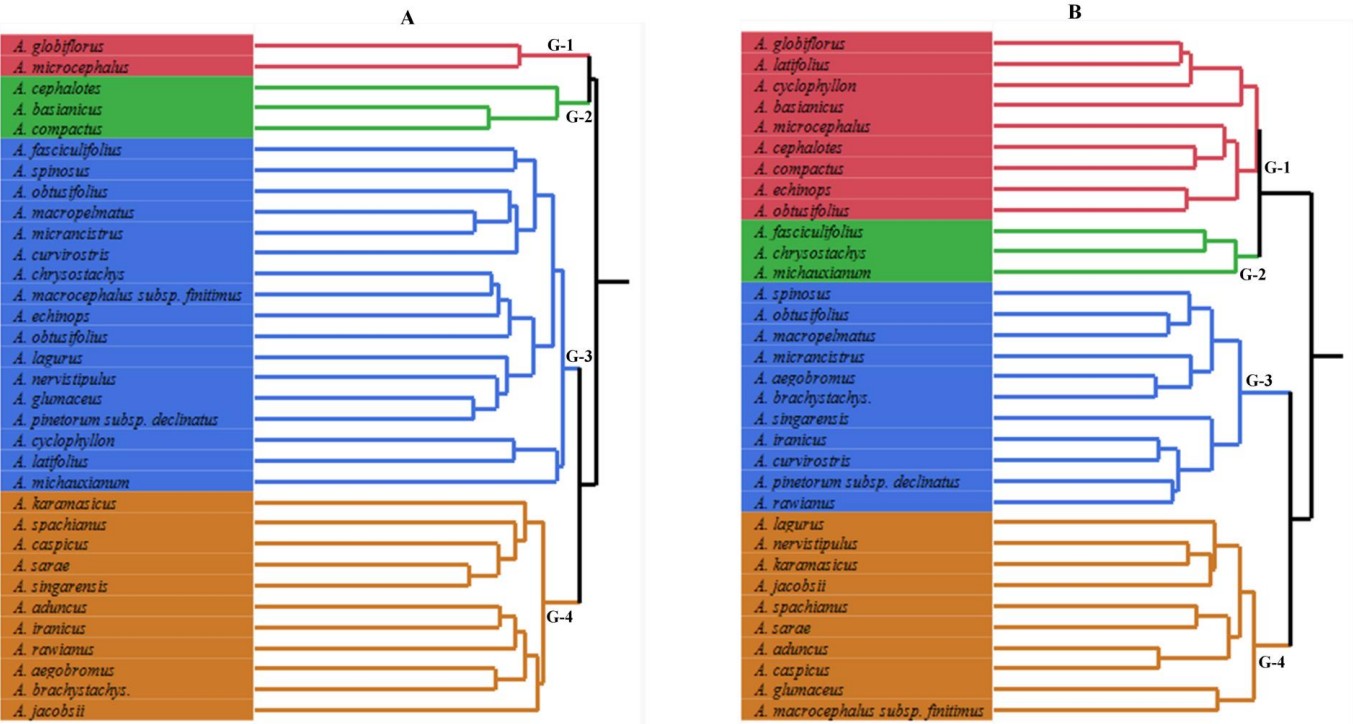

**Figure 3.** Cluster analysis of 33 *Astragalus* L. taxa based on ISSR (**A**) and CDDP (**B**) markers using the unweighted pair-group average method. Different colors denote different groups (G).

The 33 *Astragalus* taxa were divided into four clades based on the ISSR molecular data and distribution of the species on the principal coordinate plot (PCoA), as shown in Figure 4A. Ten taxa were found in the top-left quadrant of the plot in the first group. Four taxa made up the second group in the plot's top-right quadrant. Eleven taxa belonged to the third group, which is located in the bottom-left quadrant of the plot, and eight taxa belonged to the fourth group, which is located in the bottom-right quadrant of the plot.

The data from CDDP primers on *Astragalus* species revealed that all the studied taxa were divided into four distinct groups (Figure 4B). Group 1 is located in the top-left quadrant of the plot, and is composed of eight taxa. The second group, located in the top-right quadrant of the plot, included seven taxa. The third group, which is located in the bottom-left quadrant of the plot, contained nine taxa, while the fourth group, which is located in the bottom-right quadrant on the plot, consisted of eight taxa.

### 3.5. Molecular Variation Analysis of Astragalus L. Taxa

Molecular variance analyses (AMOVA) for the ISSR and CDDP data displayed the highest percentage of variation within sections. Based on AMOVA results, ISSR and CDDP detected 7.91 and 10.52% genetic variation across the 11 sections, respectively, whereas the highest percentage differences between sections were 92.09 and 89.48% for the ISSR and CDDP markers, respectively (Figure 5).

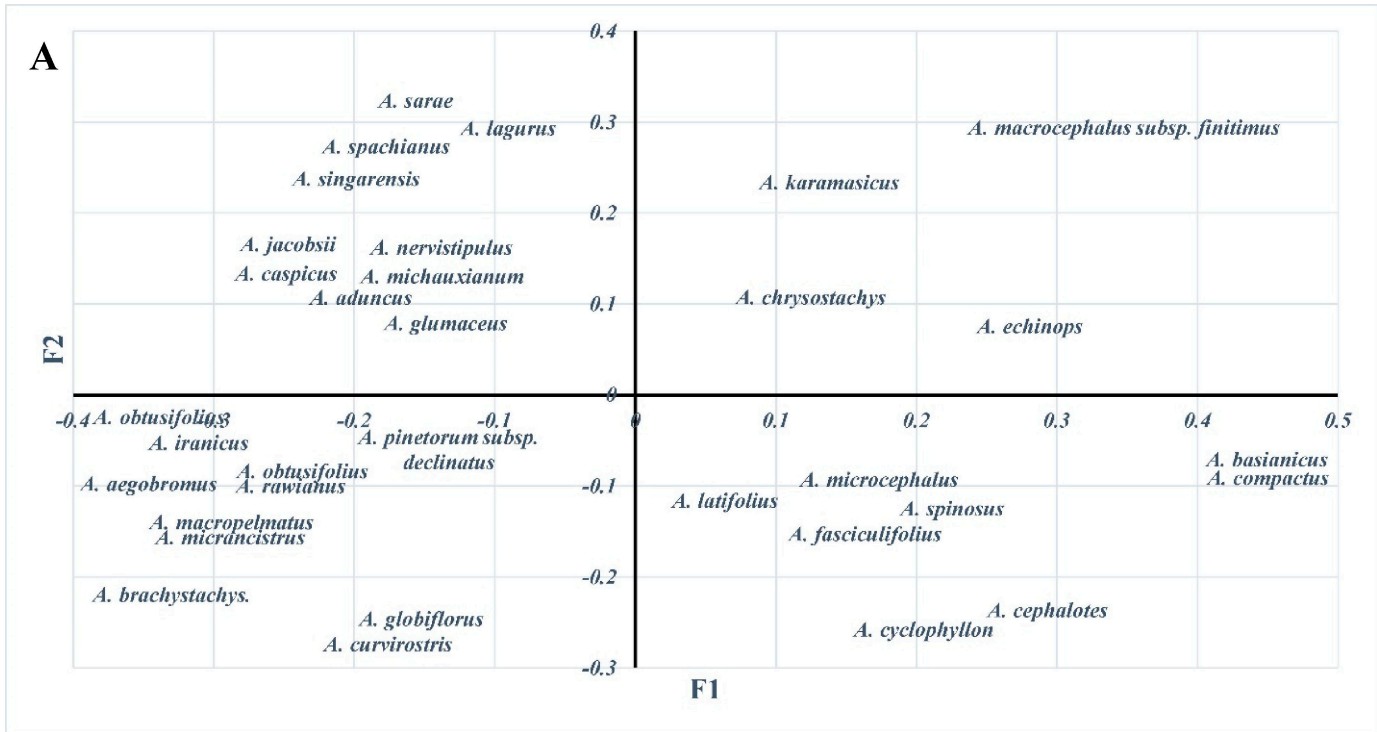

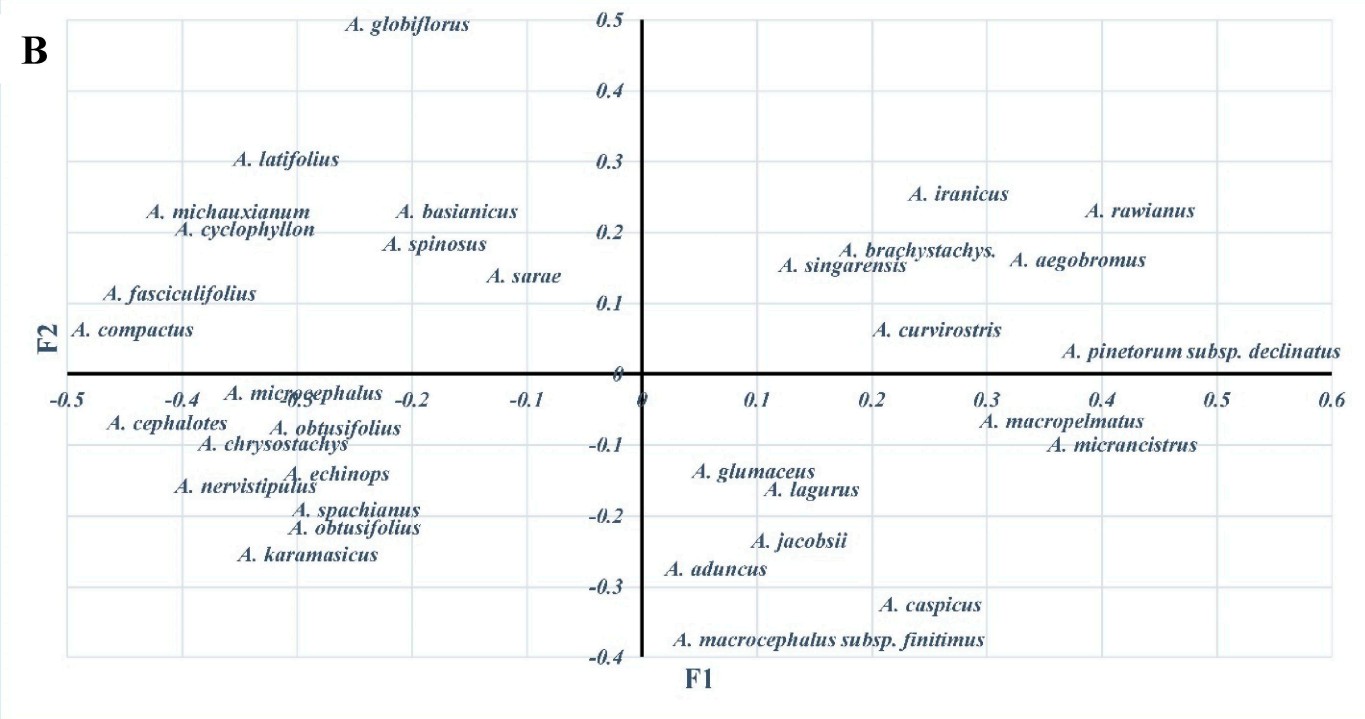

**Figure 4.** Plot based on ISSR (**A**) and CDDP (**B**) markers from principal coordinate analysis (PCoA) demonstrating genetic relatedness and variability among 33 *Astragalus* species.

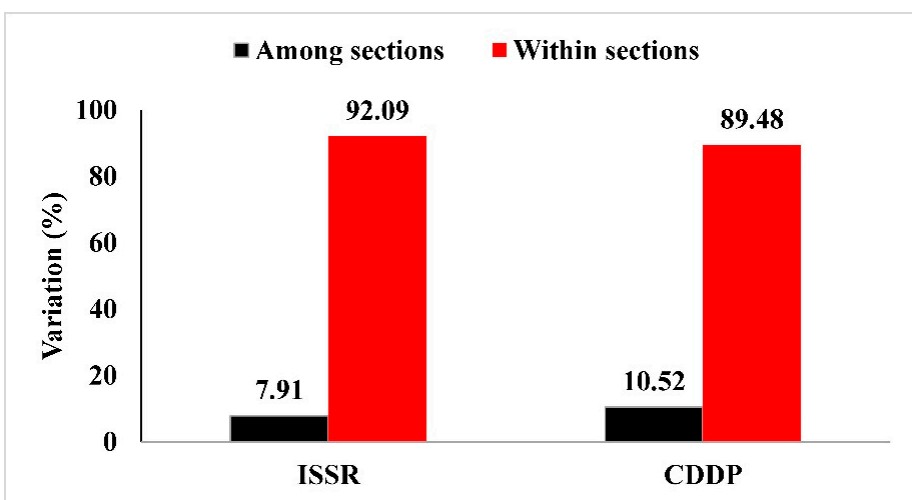

**Figure 5.** Molecular variation of eleven sections of the genus *Astragalus* among and within sections based on ISSR and CDDP molecular data sets.

## 4. Discussion

Endemism, or the frequency with which a species is found in a single geographically defined location, increases with elevation, and a substantial number of endemic species are exclusive to this habitat. Many endemic species inhabit this ecosystem, as stated recently by Lopez-Alvarado and Farris [31]. The accurate, fast, and economical identification of species has been the primary goal in taxonomy, particularly in cases of related plants, because identification based solely on phenotypic characteristics is frequently inconclusive due to the high variability within the taxa [32]. This is the most extensive phylogenetic and up-to-date analysis of *Astragalus* L. based on molecular markers and standard sequence analysis in Iraq. One of the most often used genes in molecular taxonomy or phylogenetic studies is ITS rDNA, making it a significant marker in ecological biodiversity screening [33]. A total of 33 taxa for the genus *Astragalus*, including 11 sections, were identified in various regions of Iraqi Kurdistan (Sulaimani, Erbil, Rouandwz, and Amadya), with the highest distribution for the Malacothrix section (*A. iranicus*). In Turkey, a total of 56 taxa with three sections of the genus *Astragalus* are widely distributed, including Incani (30 species), Hypoglottidei (15 species), and Dissitiflori (11 species) [34]. On the other hand, in Iran, 24 species belonging to *Astragalus* L. section Dissitiflori were identified [35]. In this study, the ITS region of rDNA was chosen to find out how the 33 species of the genus *Astragalus* found in Iraqi Kurdistan are related to each other in terms of evolution. In general, the observed species were distributed over 11 sections in this study, despite the limited taxa sampling, which is similar to the findings of Azani et al. in Iran [11]; these authors studied 210 *Astragalus* species using two different sequencing techniques: rDNA ITS and plastid DNA regions (*ycf* + *trnK/matK*). *Astragalus* L. taxa in our study had very high nucleotide sequence similarity to NCBI ITS region sequences, supporting their identification. Researchers have performed phylogenetic studies of *Astragalus* [10], using rDNA ITS and chloroplast spacers *trnD-trnT* and *trnfM-trnS1* either alone or in combination, and have found two main clades, which are consistent with our own. The taxa from the sections Proseliu, Incai, Myobroma, and Alopecias were grouped together in our study's first group, while the taxa from the sections Macrophyllium, Poterion, Rhacophorus, Onobrychium, and Adiaspastus were grouped together in the second group. The ITS region of the taxa grouped together in the same clade share a high degree of similarity, which helps to explain why they are grouped together. Furthermore, the bootstrap in this study was lowest between *A. glumaceus, A. caspicus*, and *A. lagurus* (30%), while the highest was between *A. spinous* and *A. cephalotes* (99%). Another study in Turkey reported that species-rich sections of Incani made a well-supported 99% bootstrap [34], while in the USA, the bootstrap was 88% for Neo-*Astragalus* [36]. Moreover, the highest bootstrap (100%) was between

*A. adsurgens* and *A. alpinus,* according to another study [37]. Some *Astragalus* species in this study were recorded for the first time in the NCBI, including *A. michauxianum*, *A. basianicus*, *A. sarae*, *A. singarensis*, *A. brachystachys*, and *A. rawianus*. In this respect, New World species (Hypoglottidei and Dissitiflori clusters) were also reported in Turkey [34], and in Iran, *A. lignipes* [38] and *A. taleshensis* [39]. A multi-base deletion found in the ITS1 and ITS4 regions may explain why the *Neo-Astragalus* cluster is more closely related to each subgroup. Moreover, in this study, *A. echinops* species were related to *A. obtusifolius* in the phylogenetic tree. *A. michauxianum* is an Old World species, and *A. macrocephalus* subsp. *finitimus* is in the same cluster. As a result, it is reasonable to anticipate deeper ties between these two species. According to our analysis with relatively few species, rDNA ITS sequences were used to identify phylogenetic connections for about 200 Old World *Astragalus* species [40]. In our study, the most common *Astragalus* species was *A. iranicus*, which was discovered in four regions of elevated areas ranging from 1580 to 1799 m. In comparison to our discovery, Ranjbar and Mahmoudian [41] discovered three distinct *A. iranicus* at altitudes of 1580–1799 m, indicating the evolutionary process of *Astragalus* species in our neighbouring area.

The preservation of genetic variety is one of the fundamental aims of endangered and threatened species conservation. Knowledge of genetic diversity is crucial for establishing efficient conservation management techniques [42]. The allelic richness of a species is a measure of the enrichment of genetic diversity, which is broadly used as a molecular marker to identify populations for selection, breeding, and conservation. As the number of markers and genomes rises, the data's dependability improves [43]. There may be a correlation between the number of polymorphisms detected using different markers in diversity studies and the accuracy with which genetic differences between taxa can be determined. There were 318 polymorphisms in the species due to the use of 14 ISSR markers, but in a different study, 125 bands were discovered for 40 ISSR markers in our study. Polymorphism in the genomes of the various taxa may account for these variations. CDDP markers amplify conserved regions of functional genes, while ISSR markers target microsatellite-flanked regions of the genome [19,28].

Despite a few attempts in Egypt using SCoT markers to detect genetic diversity in 10 *Astragalus* species [44], there are no published data for the CDDP marker, and this is the first report in the literature that uses this marker for genetic diversity classification of *Astragalus* species. The majority of CDDP and ISSR markers exhibited moderate average predicted heterozygozity or genetic diversity values, suggesting that these markers are useful for analyzing genetic diversity in *Astragalus* taxa and verifying the existence of considerable genetic diversity. The employed CDDP markers are more reproducible, reliable, and easy to generate than other arbitrary markers like RFLP and RAPD, since they use longer primers at conserved regions and higher annealing temperatures [45]. Furthermore, ISSRs are distributed at random throughout the genome [46]. In this regard, there is a significant possibility of polymorphism between the studied taxa, as demonstrated by our findings. Evaluation of plant germplasm diversity is a potent tool for locating high-quality breeding materials and boosting breeding efficiency [19].

In our research, a higher PIC value designated higher CDDP and ISSR marker polymorphism, which helped pick the optimal markers for genetic divergence analyses. Diversity indices quantify a population's diversity by statistically grouping members. Lower numbers imply less variability, whereas higher values indicate more [47]. The identified PIC values in this study ranged from 0.30–0.40 and 0.29–0.44, with a mean value of 0.35 and 0.37 for ISSR and CDDP markers, respectively. This displays the good capability of most primers used in taxa of *Astragalus* L., thereby providing a valuable resource for studying plant populations and identifying population genetics. The present work identified 14 ISSR markers with PIC values higher than the mean PIC value, which may prove useful for future trait mapping and tagging research involving Iraqi *Astragalus* taxa. In this regard, lower values of PIC were reported for *Astragalus* species in other studies in Iran (0.31–0.37) [16] and Romania (0.34) [48].

The mean expected heterozygosity (He) for CDDP and ISSR was 0.19 and 0.17, respectively, indicating a low level of genetic diversity across populations. The average value of Shannon's information index (I) determined from CDDP was somewhat higher than that of the ISSR marker, showing that the CDDP genome exhibits significant population variation. Three distinct *Astragalus* species populations were stated for the ISSR markers; however, only two populations were defined for the CDDP markers, similar to the work conducted on *Astragalus exscapus* using an eight-pair combination of SRAP markers [48]. In Iran, 10 ISSR markers were used to separate the *A. cyclophyllon* population into three subgroups, in support of these findings [16].

Cluster analyses are important to understand genetic diversity and simplify subsequent species mapping studies [49]. The study of the obtained dendrogram shows that the *Astragalus* taxa have significant population dispersion. The vast range of dissimilarity across sections indicates a significant degree of genetic diversity as well as a large quantity of genetic material accessible. The range of dissimilarity was higher in the ISSR marker than in the CDDP marker. A critical finding on the plot of PCoA is the applicability of both approaches (ISSR and CDDP) to the dispersion of the taxa, and the identification of the species with the best dispersion on the PCoA diagram produced from the CDDP data. Group 1 and Group 4 as well as Group 2 and Group 4 were genetically more distant, which is another intriguing characteristic of the PCoA analysis that is visible on both PCoA plots.

The main cause of the lack of correlation between the CDDP and ISSR results, as well as between them and the ITS method, is the variation in the genome of taxa, which is amplified using various techniques. The large-subunit rRNA and small-subunit ribosomal RNA genes on the chromosome are separated by spacer DNA, which is amplified by the universal barcode sequence (ITS). While ISSR markers concentrate on the genome's microsatellite-flanked regions, CDDP markers amplify the conserved regions of functional genes [19,28].

*Astragalus* taxa in Iraqi Kurdistan exhibit a distinct pattern in terms of where they grow and how their genes differ. This is most likely due to geographical isolation and allopatric speciation. Finding a new species of *Astragalus* in just one trip demonstrates the importance of studying Iraq's native plants, particularly in the Kurdistan highlands.

## 5. Conclusions

The current molecular phylogenetic study offers valuable data on the evolutionary relationships among the 33 *Astragalus* taxa that are naturally distributed in the Kurdistan region of Iraq. The MEGA 11 phylogenetic analysis of the genus *Astragalus* reveals a clear separation at the section level. The genetic relationships between *Astragalus* species were successfully determined using ISSR and CDDP markers, which demonstrated high efficiency in detecting polymorphism. Finally, we realized that the results from the present study indicated that non-traditional types of data such as sequence duplications, insertions, deletions, single-nucleotide polymorphisms, or a combination of them are valuable in elucidating the phylogenetic relationships between plant species. However, a great number of *Astragalus* species still remain to be explored in Iraq. These preliminary findings need to be verified using more representative samples. In future studies on *Astragalus* species, in addition to the rDNA ITS region, several regions from cpDNA might be useful for studying the phylogenetic relationships among sections and species of the *Astragalus* genus.

**Author Contributions:** Supervision, N.A.-r.T. and R.T.A.-R.; conceptualization, N.A.-r.T. and R.T.A.-R.; investigation, L.H.K. and N.A.-r.T.; methodology, L.H.K.; data curation, L.H.K.; formal analysis, N.A.-r.T.; writing—original draft preparation, L.H.K., R.T.A.-R. and N.A.-r.T.; writing—review and editing L.H.K., N.A.-r.T. and R.T.A.-R. All authors have read and agreed to the published version of the manuscript.

**Funding:** This research received no external funding.

**Institutional Review Board Statement:** Not applicable.

**Informed Consent Statement:** Not applicable.

**Data Availability Statement:** The article contains all data.

**Acknowledgments:** The author would like to acknowledge all the labs that contributed to this investigation.

**Conflicts of Interest:** The authors declare no conflict of interest.

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
