# Peer review of "Molecular Variation in Some Taxa of Genus Astragalus L. (Fabaceae) in the Iraqi Kurdistan Region"

_horticulturae, doi:10.3390/horticulturae9101110_

Round 1
Reviewer 1 Report
The manuscript submitted by Khal and collaborators is a very well-prepared manuscript. Materials and methods are well described, results are consistent and robust, and discussion and conclusions are linked and supported by the findings found in this study. In this point, I have minor issues:
- Table 1. Accesions numbers: How these accesions numbers were obtained? Are they herbarium vouchers o just a key used in this study? Clarify it, please.
-Lines 184, 191, and 195: Use italic these scientific names.
Author Response
Manuscript number: horticulturae-2600517
Paper title: Molecular Taxonomy for Some Taxa of Genus Astragalus L. (Fabaceae) in Iraqi Kurdistan Region
Authors: Lanja Hewa Khal, Nawroz Abdul-razzak Tahir, Rupak Tofiq Abdul-Razaq and Djshwar Dhahir Lateef
Dear Editor and Reviewer
The authors would like to thank the area editor and the reviewers for their precious time and invaluable comments. We have carefully addressed all the comments. The constructive comments/ suggestions by the reviewer are really appreciated. We have now completely revised the manuscript. All corrections in English language and updating of information are highlighted by the red lines. The corresponding changes and refinements made in the revised paper are summarized in our response below. The actual comments and questions of the reviewer are in BOLDED RC, and the author's responses are italicized AR.
Comments of Reviewer and Answer
Reviewer-1
RC1. Table 1. Accesions numbers: How these accesions numbers were obtained? Are they herbarium vouchers o just a key used in this study? Clarify it, please.
- Thank you. This section has been modified in the revised manuscript.
RC2. Lines 184, 191, and 195: Use italic these scientific names.
- Thank you. This has been modified in the revised manuscript
Best regards

Reviewer 2 Report
The manuscript is very interesting and give the valuable information to the researchers and readers. The subject of the manuscript is consistent with the scope of the Journal. The English language is fluent and easy to read. Thus, I suggested that the manuscript need to be minor revised before it is accepted by this journal.
The following specific comments are observed:
1. Abstract is too long, shorten it by at least half.
2. Line 23-24: Latin first appears to be the full name.
3. Keywords are usually 5, and the first letter should be capitalized or alphabetized.
4.The neatness of discussion must be improved. There are many paragraphs, some of which are long and some of which are short, which appear to be disorderly.
5. Conclusions is too long, shorten it by at least half.
The English language is fluent and easy to read
Author Response
Manuscript number: horticulturae-2600517
Paper title: Molecular Taxonomy for Some Taxa of Genus Astragalus L. (Fabaceae) in Iraqi Kurdistan Region
Authors: Lanja Hewa Khal, Nawroz Abdul-razzak Tahir, Rupak Tofiq Abdul-Razaq and Djshwar Dhahir Lateef
Dear Editor and Reviewer
The authors would like to thank the area editor and the reviewers for their precious time and invaluable comments. We have carefully addressed all the comments. The constructive comments/ suggestions by the reviewer are really appreciated. We have now completely revised the manuscript. All corrections in English language and updating of information are highlighted by the red lines. The corresponding changes and refinements made in the revised paper are summarized in our response below. The actual comments and questions of the reviewer are in BOLDED RC, and the author's responses are italicized AR.
Comments of Reviewer and Answer
Reviewer-2
RC1. Abstract is too long, shorten it by at least half.
AR1. Thank you. The abstract has been modified
RC2. Line 23-24: Latin first appears to be the full name.
AR2. Thank you. The abstract has been corrected
RC3. Keywords are usually 5, and the first letter should be capitalized or alphabetized.
- Thank you. The content of keywords should be written in lowercase as mentioned in the style of journal
RC4.The neatness of discussion must be improved. There are many paragraphs, some of which are long and some of which are short, which appear to be disorderly.
AR4. Thank you. The discussion has been modified and improved
RC5. Conclusions is too long, shorten it by at least half.
- Thank you. The section has been modified and improved
Best regards

Reviewer 3 Report
I state that I am not an expert on this species, however I believe that the manuscript provides some interesting new data on the evolutionary relationships among the 33 Astragalus taxa that are naturally distributed in the Kurdistan region of Iraq. The authors use three different techniques that allow them to obtain efficient data for studying plant populations.
The introduction is clear and provides comprehensive information on the state of the art of the subject matter, as well as demonstrates the importance to expand the knowledge of the taxonomic profile and genetic diversity of Astragalus species. Further, highlights the importance to have an efficient methodology to determine phylogenetic relationships among Astragalus species and to evaluate the benefit of different molecular markers such as ISSR and CDDP for investigating the taxonomical profile.
A good statistical analysis was performed.
The results are clear enough though in paragraph 3.5. (Population Structure and Dissimilarity Index in Taxa of Astragalus L.) in fact in this paragraph the exposition is unclear in some parts and figure 4 needs to be revised (you can find the notes in the manuscript).
I am surprised that the discussion in some parts is not clear, and many sentences are unrelated to each other and many times it is not clear what the authors mean. I suggest you review the discussion (you find my more detailed comments in the manuscript)
In my opinion the manuscript could be published, but it needs to be revised and deepened especially in the discussion.
I also suggest a review of English even if I'm not a native speaker.

I also suggest a review of English even if I'm not a native speaker.
Author Response
Manuscript number: horticulturae-2600517
Paper title: Molecular Taxonomy for Some Taxa of Genus Astragalus L. (Fabaceae) in Iraqi Kurdistan Region
Authors: Lanja Hewa Khal, Nawroz Abdul-razzak Tahir, Rupak Tofiq Abdul-Razaq and Djshwar Dhahir Lateef
Dear Editor and Reviewer
The authors would like to thank the area editor and the reviewers for their precious time and invaluable comments. We have carefully addressed all the comments. The constructive comments/ suggestions by the reviewer are really appreciated. We have now completely revised the manuscript. All corrections in English language and updating of information are highlighted by the red lines. The corresponding changes and refinements made in the revised paper are summarized in our response below. The actual comments and questions of the reviewer are in BOLDED RC, and the author's responses are italicized AR.
Comments of Reviewer and Answer
Reviewer-3
RC1. The results are clear enough though in paragraph 3.5. (Population Structure and Dissimilarity Index in Taxa of Astragalus L.) in fact in this paragraph the exposition is unclear in some parts and figure 4 needs to be revised (you can find the notes in the manuscript).
AR1. Thank you. The section (Population Structure and Dissimilarity Index in Taxa of Astragalus L) has been deleted because the fourth reviewer requested it.
RC2. I am surprised that the discussion in some parts is not clear, and many sentences are unrelated to each other and many times it is not clear what the authors mean. I suggest you review the discussion (you find my more detailed comments in the manuscript)
AR2. Thank you. The discussion has been improved and organized
RC3. In my opinion the manuscript could be published, but it needs to be revised and deepened especially in the discussion.
AR3. Thank you. The revision has been done
RC4. I also suggest a review of English even if I'm not a native speaker.
AR4. Thank you. The manuscript has been checked at the level of grammar and spelling and the errors have been corrected
RC5. Comments of PDF file
AR5. Thank you. All comments of PDF file requested by the reviewer have been corrected in the revised version
Best regards

Reviewer 4 Report
The study is aimed at taxonomic analysis of Astragalus species from Kurdistan using molecular methods. The authors studied sequences of nrITS region and conducted a fragment DNA analysis using a set of ISSR and CCDP markers. The authors investigated 33 species of Astragalus, distributed in several Kurdistan region of Iraq, i.e. Sulaimaniyah (MSU), Erbil (FAR), Kirkuk (FKI), Rouandwz (MRO), and Amadya provinces (MAM).
The work is rather large and includes many methods of analysis. However, as for methodological approaches, the present study has not been developed properly. I would like to stop at several important moments.
1. The authors did not explain how they identified the samples by morphological features, which literary sources were used for this. It is unclear which taxonomic system of the genus Astragalus the authors adhere to. Taxonomical names are given without author names.
2. The selection of species is limited only by geographical principle. Related species from the same sections, but native to other countries, were not included. There are also no outgroups. To obtain a more adequate phylogenetic reconstruction, it is necessary to include a representative set of Astragalus species (representing at least all subgenera and main branches) and several outgroups.
3. Sections of the genus Astragalus are not shown on the tree. It is unclear whether the sectional division of the genus corresponds to the topology of the phylogenetic tree or not.
4. As far as I could understand, only one representative specimen from each studied species was selected for nrITS sequencing, however most of species (20 of 33) are known from two or more regions. To evaluate the genetic distance between studied species of Astragalus, the authors applied a distance based method such as Neighbor-joining tree reconstruction. The analysis demonstrated that studied specimens are more or less separated from each other by ITS, but some branches have a very low bootstrap support, and the position of these species is uncertain. It is still very early to draw a conclusion about the isolation of the species according to nrITS marker, since each species is represented by a single specimen and intraspecific variability is unknown. Despite some shortcomings, the analysis of Neighbor-joining by ntITS sequences is a suitable method for assessing the degree of genetic differentiation of species in this work.
5. The second part of the study deals with fragment DNA analysis using ISSR and CDDP markers. Methods of analysis of ISSR and CDDP data were the following: 3.2 and 3.3) analysis of polymorphism and genetic diversity of ISSR and CDDP primers using several parameters (NPB: number of polymorphic bands, PIC: polymorphic information content, Na: number of different alleles, Ne: number of effective alleles, I: Shannon’s information index, He: expected heterozygosity, uHe: unbiased expected heterozygosity); 3.4) Cluster Analysis of Astragalus L. species; 3.5) Population Structure and Dissimilarity Index in Taxa of Astragalus L. using the STRUCTURE program and Molecular variance analyses (AMOVA). From these methods, Cluster analysis is more adequate for such data. It allows to evaluate genetic distance between studied specimens. Another approaches that can be applied to such binary data is a Principal Coordinate Analysis based on Jaccard distance or similar metric. I would recommend the authors to try this method. Analysis of population structure using the STRUCTURE is unsuitable for such data, because each studied specimen represents a separate species. The STRUCTURE software is designed to analyze population samples of one species or several closely related species. This method assumes a Hardy-Weinberg equilibrium within genetic clusters, which does not correspond to the data set used in this study. Molecular variance analyses (AMOVA) is applicable to the studied dataset. As for the analysis of polymorphism and genetic diversity indices, I also believe that these indices are applicable to population samples, and not to single specimens of different species. Moreover, when using dominant markers (as ISSR and CDDP), you can only indirectly estimate such parameters of genetic diversity as the number of different alleles, the number of effective alleles, expected heterozygosity, etc.
6. In general, I would like to emphasize that in order to evaluate the effectiveness of certain markers for species separation, you need to compare interspecific and intraspecific variability (i.e. interspecific and intraspecific variability). Having one sample of each species, you can't do that.
7. In conclusion, I would recommend the authors to significantly revise the article in the following directions: 1) include a more representative set of samples of Astragalus and outgroups in the phylogenetic analysis by nrITS (for example, using GenBank data); apply subgenera and sections to the phylogenetic tree; 2) exclude the analysis in the Structure program from the article; 3) perhaps, exclude the indices of genetic diversity also; 4) AMOVA analysis may be included; 5) you can also include cluster analysis by ISSR and CDDP markers and add another method of analyzing the same data (Neighbor-Joining or Principal Coordinate Analysis); apply species and sections to dendrograms; 5) compare the results of nrITS and fragment analysis in taxonomic terms.
Several comments are made in the pdf-file.

Author Response
Manuscript number: horticulturae-2600517
Paper title: Molecular Taxonomy for Some Taxa of Genus Astragalus L. (Fabaceae) in Iraqi Kurdistan Region
Authors: Lanja Hewa Khal, Nawroz Abdul-razzak Tahir, Rupak Tofiq Abdul-Razaq and Djshwar Dhahir Lateef
Dear Editor and Reviewer
The authors would like to thank the area editor and the reviewers for their precious time and invaluable comments. We have carefully addressed all the comments. The constructive comments/ suggestions by the reviewer are really appreciated. We have now completely revised the manuscript. All corrections in English language and updating of information are highlighted by the red lines. The corresponding changes and refinements made in the revised paper are summarized in our response below. The actual comments and questions of the reviewer are in BOLDED RC, and the author's responses are italicized AR.
Comments of Reviewer and Answer
Reviewer-4
RC1.The authors did not explain how they identified the samples by morphological features, which literary sources were used for this. It is unclear which taxonomic system of the genus Astragalus the authors adhere to. Taxonomical names are given without author names.
AR1. Thank you. The detail of the identification of taxa by morphological method has been added to the revised manuscript
RC2. The selection of species is limited only by geographical principle. Related species from the same sections, but native to other countries, were not included. There are also no outgroups. To obtain a more adequate phylogenetic reconstruction, it is necessary to include a representative set of Astragalus species (representing at least all subgenera and main branches) and several outgroups.
AR2. I'm grateful. Instead of focusing on different genotypes of a single taxon, the goal of this study is to identify the molecular signature or variation among various Astragalus taxa. Since we have 33 taxa and compare each taxa with the NCBI database, it is not necessary to compare our ITS findings with the other databases on the NCBI because this would result in 33 dendrograms, which is unacceptable for the journal.
RC3. Sections of the genus Astragalus are not shown on the tree. It is unclear whether the sectional division of the genus corresponds to the topology of the phylogenetic tree or not.
AR3. Thank you. The sections are shown on the phylogenetic tree (Figure 2)
RC4. As far as I could understand, only one representative specimen from each studied species was selected for nrITS sequencing, however most of species (20 of 33) are known from two or more regions. To evaluate the genetic distance between studied species of Astragalus, the authors applied a distance based method such as Neighbor-joining tree reconstruction. The analysis demonstrated that studied specimens are more or less separated from each other by ITS, but some branches have a very low bootstrap support, and the position of these species is uncertain. It is still very early to draw a conclusion about the isolation of the species according to nrITS marker, since each species is represented by a single specimen and intraspecific variability is unknown. Despite some shortcomings, the analysis of Neighbor-joining by ntITS sequences is a suitable method for assessing the degree of genetic differentiation of species in this work.
AR4. Thank you. A few taxa are available in two or more areas. We examined every sample gathered from every location, and the ITS results were the same everywhere. Therefore, as I mentioned in the text above, the purpose of this study is not to investigate various genotypes or accessions of a particular taxon; rather, it is to investigate the molecular variation of various taxa collected from various locations. Accordingly, it is not necessary to use all of the samples that were collected.
RC5. In conclusion, I would recommend the authors to significantly revise the article in the following directions: 1) include a more representative set of samples of Astragalus and outgroups in the phylogenetic analysis by nrITS (for example, using GenBank data); apply subgenera and sections to the phylogenetic tree; 2) exclude the analysis in the Structure program from the article; 3) perhaps, exclude the indices of genetic diversity also; 4) AMOVA analysis may be included; 5) you can also include cluster analysis by ISSR and CDDP markers and add another method of analyzing the same data (Neighbor-Joining or Principal Coordinate Analysis); apply species and sections to dendrograms; 5) compare the results of nrITS and fragment analysis in
AR5. Thank you. All requested corrections of this section have been done in the revised manuscript
RC6. Comments of PDF file
AR6. Thank you. All comments of PDF file requested by the reviewer have been corrected in the revised version
Best regards

Round 2
Reviewer 4 Report
Dear Editors and Authors, my new comments are marked in red.
Manuscript number: horticulturae-2600517
Authors: Lanja Hewa Khal, Nawroz Abdul-razzak Tahir, Rupak Tofiq Abdul-Razaq and Djshwar Dhahir Lateef
Dear Editor and Reviewer
The authors would like to thank the area editor and the reviewers for their precious time and invaluable comments. We have carefully addressed all the comments. The constructive comments/ suggestions by the reviewer are really appreciated. We have now completely revised the manuscript. All corrections in English language and updating of information are highlighted by the red lines. The corresponding changes and refinements made in the revised paper are summarized in our response below. The actual comments and questions of the reviewer are in BOLDED RC, and the author's responses are italicized AR.
Comments of Reviewer and Answer
Reviewer-4
RC1.The authors did not explain how they identified the samples by morphological features, which literary sources were used for this. It is unclear which taxonomic system of the genus Astragalus the authors adhere to. Taxonomical names are given without author names.
AR1. Thank you. The detail of the identification of taxa by morphological method has been added to the revised manuscript – Partially corrected. The authors did not answer, what taxonomic system of the genus Astragalus they used.
RC2. The selection of species is limited only by geographical principle. Related species from the same sections, but native to other countries, were not included. There are also no outgroups. To obtain a more adequate phylogenetic reconstruction, it is necessary to include a representative set of Astragalus species (representing at least all subgenera and main branches) and several outgroups.
AR2. I'm grateful. Instead of focusing on different genotypes of a single taxon, the goal of this study is to identify the molecular signature or variation among various Astragalus taxa. Since we have 33 taxa and compare each taxa with the NCBI database, it is not necessary to compare our ITS findings with the other databases on the NCBI because this would result in 33 dendrograms, which is unacceptable for the journal. – The authors answered another question. I asked why they did not include other groups of Astragalus and outgroups in phylogenetic analysis.
RC3. Sections of the genus Astragalus are not shown on the tree. It is unclear whether the sectional division of the genus corresponds to the topology of the phylogenetic tree or not.
AR3. Thank you. The sections are shown on the phylogenetic tree (Figure 2). – Corrected.
RC4. As far as I could understand, only one representative specimen from each studied species was selected for nrITS sequencing, however most of species (20 of 33) are known from two or more regions. To evaluate the genetic distance between studied species of Astragalus, the authors applied a distance based method such as Neighbor-joining tree reconstruction. The analysis demonstrated that studied specimens are more or less separated from each other by ITS, but some branches have a very low bootstrap support, and the position of these species is uncertain. It is still very early to draw a conclusion about the isolation of the species according to nrITS marker, since each species is represented by a single specimen and intraspecific variability is unknown. Despite some shortcomings, the analysis of Neighbor-joining by ntITS sequences is a suitable method for assessing the degree of genetic differentiation of species in this work.
AR4. Thank you. A few taxa are available in two or more areas. We examined every sample gathered from every location, and the ITS results were the same everywhere. Therefore, as I mentioned in the text above, the purpose of this study is not to investigate various genotypes or accessions of a particular taxon; rather, it is to investigate the molecular variation of various taxa collected from various locations. Accordingly, it is not necessary to use all of the samples that were collected. – If ITS sequences of the same species from multiple locations were identical, why didn't you publish each sequence under a separate accession number, regardless of whether they were identical or not?
RC5. In conclusion, I would recommend the authors to significantly revise the article in the following directions: 1) include a more representative set of samples of Astragalus and outgroups in the phylogenetic analysis by nrITS (for example, using GenBank data); apply subgenera and sections to the phylogenetic tree; 2) exclude the analysis in the Structure program from the article; 3) perhaps, exclude the indices of genetic diversity also; 4) AMOVA analysis may be included; 5) you can also include cluster analysis by ISSR and CDDP markers and add another method of analyzing the same data (Neighbor-Joining or Principal Coordinate Analysis); apply species and sections to dendrograms; 5) compare the results of nrITS and fragment analysis in
AR5. Thank you. All requested corrections of this section have been done in the revised manuscript – The recommendation 1) include a more representative set of samples of Astragalus and outgroups in the phylogenetic analysis by nrITS (for example, using GenBank data) – not corrected;
RC6. Comments of PDF file
AR6. Thank you. All comments of PDF file requested by the reviewer have been corrected in the revised version
Page 1, abstract: “Astragalus L. is one of the main genera of blossoming trees” – – not corrected. It is known that Astragalus includes herbs and small shrubs.
Some other items to comment:
Page 2. “Thus, the current study intends to use nucleotide sequences of ITS to identify 33 Astragalus species in Kurdistan, Iraq. Furthermore, to determine phylogenetic relationships among Astragalus species and to evaluate the benefit of molecular markers for investigating the taxonomical profile and population content of Astragalus species, ISSR and CDDP markers were examined. A molecular phylogenetic investigation of Astragalus species in Iraqi Kurdistan would be a significant addition to understanding the evolutionary link between neighboring area and the distribution of Astragalus species”. – Here and later in the text the authors write about phylogenetic relationships and phylogenetic investigation. But phylogenetic analysis requires the inclusion of external groups.
Page 11. “In general, the observed species were distributed over 11 sections in this study despite the limited taxa sampling, similar to the findings conducted in Iran by Azani et al. [11], who studied 210 Astragalus species using two different sequencing techniques: rDNA ITS and plastid DNA regions (ycf+ trnK/matK), demonstrating the originality of this work in selecting and distributing these species in the studied area” – the logic of this sentence is unclear.
Figures 2, 3 and 4.
It seems that there is no correlation between the results obtained using CDDP and ISSR markers. Perhaps the observed variability reflects individual (rather than taxonomic) variability. To check the significance of these markers for defining the boundaries of species, you need to:
1. Repeat all analyses twice to confirm reproducibility of bands.
2. Include two or more specimens of each species to differentiate inter- and intraspecific variability.
In the present form, you cannot say that the CDDP and ISSR markers are useful for distinguishing species in the study group. I recommend emphasizing in the discussion the lack of correlation between the two types of fragment analysis markers (CDDP and ISSR), as well as between them and the nrITS marker.
Concerning the results of fragment DNA analysis (i.e. CDDP and ISSR), I would also recommend adding the phrase that these are still very preliminary results that should be treated with caution and that require testing on more representative material.
Page 11-12. “The taxa from the sections Proseliu, Incai, Myobroma, and Alopecias were grouped together in our study's first group, while the taxa from the sections Macrophyllium, Poterion, Rha-cophorus, Onobrychium, and Adiaspastus were grouped together in the second group. Based on these findings, it is reasonable to assume that the geographical distributions of different Astragalus species are quite variable, ranging from species that are endemic to a single region to species that are widespread across an entire continent. – Why?
Page 12. “…comparable to our analysis with relatively higher species.” – “relatively higher species” is not a good term.
Page 12. “There were 318 polymorphisms in the species due to the use of 14 ISSR markers, but in a different study, 125 bands were discovered for 40 ISSR markers in our study” – What study do you discuss? Please, give the reference.

Author Response
Manuscript number: horticulturae-2600517-R2
Paper title: Molecular variation of Some Taxa of Genus Astragalus L. (Fabaceae) in the Iraqi Kurdistan Region
Authors: Lanja Hewa Khal, Nawroz Abdul-razzak Tahir and Rupak Tofiq Abdul-Razaq
Dear Editor and Reviewer
The authors would like to thank the area editor and the reviewers for their precious time and invaluable comments. We have carefully addressed all the comments. The constructive comments/ suggestions by the reviewer are really appreciated. We have now completely revised the manuscript. All corrections in English language and updating of information are highlighted by the red lines. The corresponding changes and refinements made in the revised paper are summarized in our response below. The actual comments and questions of the reviewer are in BOLDED RC, and the author's responses are italicized AR.
Comments of Reviewer and Answer
Reviewer-4
RC1.The authors did not explain how they identified the samples by morphological features, which literary sources were used for this. It is unclear which taxonomic system of the genus Astragalus the authors adhere to. Taxonomical names are given without author names.
AR1. Thank you. The flora of Iraq, Iran, and Turkey (morphological characters) have been used as the indicator for the classification of different taxa
RC2. The selection of species is limited only by geographical principle. Related species from the same sections, but native to other countries, were not included. There are also no outgroups. To obtain a more adequate phylogenetic reconstruction, it is necessary to include a representative set of Astragalus species (representing at least all subgenera and main branches) and several outgroups.
AR2. I'm grateful. Instead of focusing on different genotypes of a single taxon, the goal of this study is to determine the relationship among 33 Astragalus taxa. The objective of this study is not to comparing the different locations of the sampling. Since we have 33 taxa and compare each taxa with the NCBI database, it is not necessary to compare our ITS findings with the other databases on the NCBI because this would result in 33 dendrograms, which is unacceptable for the journal. This section has been updated in the revised manuscript.
RC3. Sections of the genus Astragalus are not shown on the tree. It is unclear whether the sectional division of the genus corresponds to the topology of the phylogenetic tree or not.
AR3. Thank you. The sections are shown on the phylogenetic tree (Figure 2)
RC4. As far as I could understand, only one representative specimen from each studied species was selected for nrITS sequencing, however most of species (20 of 33) are known from two or more regions. To evaluate the genetic distance between studied species of Astragalus, the authors applied a distance based method such as Neighbor-joining tree reconstruction. The analysis demonstrated that studied specimens are more or less separated from each other by ITS, but some branches have a very low bootstrap support, and the position of these species is uncertain. It is still very early to draw a conclusion about the isolation of the species according to nrITS marker, since each species is represented by a single specimen and intraspecific variability is unknown. Despite some shortcomings, the analysis of Neighbor-joining by ntITS sequences is a suitable method for assessing the degree of genetic differentiation of species in this work.
AR4. Thank you. Instead of looking into different genotypes or accessions of a specific taxon, the aim of this study is to look into the molecular diversity of various taxa. Therefore, not all of the samples that were gathered have to be used.
RC5. In conclusion, I would recommend the authors to significantly revise the article in the following directions: 1) include a more representative set of samples of Astragalus and outgroups in the phylogenetic analysis by nrITS (for example, using GenBank data); apply subgenera and sections to the phylogenetic tree; 2) exclude the analysis in the Structure program from the article; 3) perhaps, exclude the indices of genetic diversity also; 4) AMOVA analysis may be included; 5) you can also include cluster analysis by ISSR and CDDP markers and add another method of analyzing the same data (Neighbor-Joining or Principal Coordinate Analysis); apply species and sections to dendrograms; 5) compare the results of nrITS and fragment analysis in
AR5. Thank you. All requested corrections of this section have been updated in the revised manuscript
RC6. Page 1, abstract: “Astragalus L. is one of the main genera of blossoming trees” – – not corrected. It is known that Astragalus includes herbs and small shrubs.
AR6. Thank you. This section has been corrected
RC7. Page 2. “Thus, the current study intends to use nucleotide sequences of ITS to identify 33 Astragalus species in Kurdistan, Iraq. Furthermore, to determine phylogenetic relationships among Astragalus species and to evaluate the benefit of molecular markers for investigating the taxonomical profile and population content of Astragalus species, ISSR and CDDP markers were examined. A molecular phylogenetic investigation of Astragalus species in Iraqi Kurdistan would be a significant addition to understanding the evolutionary link between neighboring area and the distribution of Astragalus species”. – Here and later in the text the authors write about phylogenetic relationships and phylogenetic investigation. But phylogenetic analysis requires the inclusion of external groups.
AR7. Thank you. This section has been revised
RC8. Page 11. “In general, the observed species were distributed over 11 sections in this study despite the limited taxa sampling, similar to the findings conducted in Iran by Azani et al. [11], who studied 210 Astragalus species using two different sequencing techniques: rDNA ITS and plastid DNA regions (ycf+ trnK/matK), demonstrating the originality of this work in selecting and distributing these species in the studied area” – the logic of this sentence is unclear
AR8. Thank you. This section has been rewritten
RC9. Figures 2, 3 and 4. It seems that there is no correlation between the results obtained using CDDP and ISSR markers. Perhaps the observed variability reflects individual (rather than taxonomic) variability. To check the significance of these markers for defining the boundaries of species, you need to: 1. Repeat all analyses twice to confirm reproducibility of bands. 2. Include two or more specimens of each species to differentiate inter- and intraspecific variability. In the present form, you cannot say that the CDDP and ISSR markers are useful for distinguishing species in the study group. I recommend emphasizing in the discussion the lack of correlation between the two types of fragment analysis markers (CDDP and ISSR), as well as between them and the nrITS marker. Concerning the results of fragment DNA analysis (i.e. CDDP and ISSR), I would also recommend adding the phrase that these are still very preliminary results that should be treated with caution and that require testing on more representative material.
AR9. Thank you. The same outcomes are obtained after repeating the data analysis three times. It is not necessary to collect two specimens from each taxon because the goal of this study was to ascertain the relationships among 33 taxa, not to compare the molecular profiles of various specimens collected in various places. Updated in this section are all suggested comments.
RC10. Page 11-12. “The taxa from the sections Proseliu, Incai, Myobroma, and Alopecias were grouped together in our study's first group, while the taxa from the sections Macrophyllium, Poterion, Rhacophorus, Onobrychium, and Adiaspastus were grouped together in the second group. Based on these findings, it is reasonable to assume that the geographical distributions of different Astragalus species are quite variable, ranging from species that are endemic to a single region to species that are widespread across an entire continent. – Why
AR10. Thank you. This section has been modified in the updated version
RC11. Page 12. “…comparable to our analysis with relatively higher species.” – “relatively higher species” is not a good term.
AR11. Thank you. This phrase has been modified in the updated version
RC12. Page 12. “There were 318 polymorphisms in the species due to the use of 14 ISSR markers, but in a different study, 125 bands were discovered for 40 ISSR markers in our study” – What study do you discuss? Please, give the reference
AR12. Thank you. The references have been added to the revised version
Best regards
